

# Gazing left, gazing right: exploring a spatial bias in social attention

Mario Dalmaso[1], Giacomo Fedrigo[1] and Michele Vicovaro[2]

[1] Department of Developmental and Social Psychology, University of Padua, Padua, Italy
[2] Department of General Psychology, University of Padua, Padua, Italy

## ABSTRACT

Faces oriented rightwards are sometimes perceived as more dominant than faces oriented leftwards. In this study, we explored whether faces oriented rightwards can also elicit increased attentional orienting. Participants completed a discrimination task in which they were asked to discriminate, by means of a keypress, a peripheral target. At the same time, a task-irrelevant face oriented leftwards or rightwards appeared at the centre of the screen. The results showed that, while for faces oriented rightwards targets appearing on the right were responded to faster as compared to targets appearing on the left, for faces oriented leftwards no differences emerged between left and right targets. Furthermore, we also found a negative correlation between the magnitude of the orienting response elicited by the faces oriented leftwards and the level of conservatism of the participants. Overall, these findings provide evidence for the existence of a spatial bias reflected in social orienting.

## INTRODUCTION

It is well established that humans can orient visual attention in response to spatial signals coming from others, a phenomenon that is often referred to as 'social attention'. For instance, attentional shifts can be elicited by the walking direction of a model (*e.g.*, *Dalmaso, 2023*; *Troje & Westhoff, 2006*; *Liu et al., 2021*) or by the orientation of a static body within space (*e.g.*, *Azarian, Esser & Peterson, 2016*; *Azarian et al., 2017*). Attentional shifts can also be elicited by pointing gestures and fingers (*e.g.*, *Ariga & Watanabe, 2009*; *Gregory & Hodgson, 2012*; *Dalmaso et al., 2015*). Nevertheless, in everyday social interaction, the most used and effective spatial signals come from the upper parts of the body (*i.e.,* the head and the gaze), which provide a more direct and unambiguous source of information concerning where others are paying attention. Head-mediated and gaze-mediated orienting of attention are at the heart of a fruitful research vein that ranges from studies conducted in animals (*e.g.*, *Shepherd, 2010*; *Zeiträg, Jensen & Osvath, 2022*), infants (*e.g.*, *Farroni et al., 2004*; *Guillon et al., 2014*), adults (*e.g.*, *McKay et al., 2021*; *Dalmaso, 2022*), up to the most recent contexts of human–robot interaction (*e.g.*, *Chevalier et al., 2020*). Social attention can be considered a building block of social relationships and it may also be involved in social development (*e.g.*, *Guillon et al., 2014*; *Dalmaso, Castelli & Galfano, 2021*). Its study

Corresponding author
Mario Dalmaso,
mario.dalmaso@unipd.it

is therefore of great interest as it may provide important insights about some fundamental mechanisms involved in everyday social interactions.

From an experimental perspective, social attention has been widely studied through the adoption of spatial cueing tasks in which, typically, a task-irrelevant social stimulus (*e.g.*, a face oriented left or right), presented at the centre of the screen, preceded the appearance of a peripheral target which required a behavioural response (*e.g.*, a key press). In general, a behavioural benefit (*e.g.*, smaller latencies and a greater accuracy) was observed on trials in which the target appeared in the same spatial position indicated by the social cue (*i.e.*, a spatially-congruent trial) than in a different position (*i.e.*, a spatially-incongruent trial; see, *e.g.*, *Friesen & Kingstone, 1998*; *Langton & Bruce, 1999*; *Cooney, Brady & Ryan, 2017*), reflecting a spatial cueing effect.

Researchers in recent decades have provided increasing evidence showing that this form of social orienting can be shaped by several social variables characterising the observer on the one hand, the cueing face on the other hand, and their relationship (*e.g.*, *Dalmaso, Castelli & Galfano, 2020*). Of interest to the present study, some works have reported that a greater orienting response (*i.e.*, a greater behavioural benefit on spatially-congruent trials than on spatially-incongruent trials) can be observed in response to faces perceived as higher in the social hierarchy than faces occupying lower positions. In these works, differences in the hierarchy were operationalised both: (1) at a perceptual level by varying the degree of physical dominance –namely, some faces were artificially masculinised (*i.e.*, dominant individuals; *e.g.*, they had heavier brow-ridges and larger jaws) or feminised (*i.e.*, subordinate individuals; *e.g.*, they had smaller brows, jaws and noses; see *Jones et al., 2010*; *Jones et al., 2011*; *Ohlsen, van Zoest & van Vugt, 2013*); and (2) at a more abstract level, by varying the information associated with different face identities—namely, some faces were described as belonging to high-status individuals, such as university teachers, whereas other faces as belonging to low-status individuals, such as workers (*e.g.*, *Dalmaso et al., 2012*; *Dalmaso et al., 2014*; *Ciardo et al., 2021*).

The literature on social cognition also showed that differences in the perception of dominance can be reported by simply varying the direction of the face (*e.g.*, *Suitner, Maass & Ronconi, 2017*; *Mendonça, Garrido & Semin, 2020a*). For instance, in *Suitner, Maass & Ronconi (2017)*, participants were presented with pictures of faces oriented leftwards or rightwards and were asked to rate the face stimuli on six-point scales at the extremes of which there were two opposing adjectives (*i.e.*, active–passive, dynamic–not dynamic, dominant–submissive and strong–weak). This was aimed at evaluating, for the two types of facial stimuli, the overall perceived level of 'agency', which can be broadly described as the ability of an individual to have an influence on others (*e.g.*, *Abele & Wojciszke, 2007*; *Hitlin & Elder, 2007*). The results reported by *Suitner, Maass & Ronconi (2017)* showed higher levels of perceived agency for faces oriented rightwards than leftwards. Similar results have been reported in other social contexts. For instance, a goal in a football match was judged as more powerful and faster, or a film scene was judged as more violent and harmful, when these actions were presented from left to right than from right to left (*Maass, Pagani & Berta, 2007*). Indirect evidence of this bias can also be found in art: it has been observed that faces portrayed in paintings produced across different centuries were preferably depicted

from left to right in the case of male individuals and from right to left in the case of female individuals (*e.g.*, *Chatterjee, 2002*). Additionally, paintings by Leonardo da Vinci representing individuals facing right were judged to be more 'potent' than individuals facing left (*Benjafield & Segalowitz, 1995*). Despite all this converging evidence, the nature of this kind of 'spatial agency bias' (for a review, see also *Suitner, Maass & Ronconi, 2017*) is still debated. A possible explanation can be found by considering cultural habits such as reading/writing direction. Reading and writing are two activities that occupy a considerable time of our everyday life, and that are generally made following a constant direction, such as from left to right in languages like Italian or English. Moreover, in these two languages, the same left–right direction flow is also reflected at the syntactic level in which the subject (the executor of an action) appears on the left side of the object (the receiver of such an action; see *Maass, Suitner & Nadhmi, 2014*). In turn, these linguistic properties would shape the way individuals would think about actions and social relationships, with the beginning/executor of an action that would be hypothetically represented on the left side of the space, and the end/receiver of that action that would be represented on the right side of the space. It is important to note that in cultures where reading/writing goes from right to left, the direction of this spatial bias can be inverted (see, *e.g.*, *Maass et al., 2009*; *Smith & Elias, 2013*), a result that reinforces the role of cultural aspects in driving this phenomenon.

In addition to the mechanism associated with person perception, faces oriented leftwards or rightwards can also influence the mechanisms that support social attention. This was reported in a recent study (*Mendonça, Garrido & Semin, 2020b*) in which participants were asked to discriminate a peripheral target presented alongside a task-irrelevant central face oriented leftwards or rightwards. The main results showed a greater orienting response (*i.e.*, a greater behavioural benefit on spatially-congruent trials than on spatially-incongruent trials) for faces oriented rightwards than leftwards, in line with the spatial agency bias described above. Overall, face orientation seems capable of shaping different mechanisms related to both social perception and attentional orienting. Nevertheless, because the study of *Mendonça, Garrido & Semin (2020b)* represents, so far, the only attempt to investigate the possible impact of this spatial bias on social attention, we deemed it worthwhile to further explore this topic.

## The present study

The purpose of this work was twofold. First, we wanted to replicate the main finding reported by *Mendonça, Garrido & Semin (2020b)*, according to which a stronger social attentional orienting can be observed for faces oriented rightwards than leftwards. Second, we wanted to explore the possible link between this peculiar phenomenon of social orienting and dominance. For this reason, we also collected a measure concerning the perceived levels of dominance associated with the facial stimuli used in the spatial cueing task, assuming that higher levels of dominance should have emerged for faces oriented rightwards than leftwards, in line with previous studies (*e.g.*, *Suitner, Maass & Ronconi, 2017*; *Mendonça, Garrido & Semin, 2020a*). In addition, we also collected a measure concerning the level of liberalism and conservatism of each participant. Indeed, there is evidence showing that individuals with higher levels of conservatism would tend to disfavour facial stimuli

characterised by lower levels of dominance (see, *e.g., Laustsen & Petersen, 2015; Laustsen & Petersen, 2016; Olivola, Tingley & Todorov, 2018*; see also *Liuzza et al., 2011*). Hence, we also explored whether the level of liberalism and conservatism was a factor capable of influencing the orienting response elicited by two types of faces which were expected to be characterised by a different level of perceived dominance.

## MATERIALS & METHODS

We report how we determined our sample size, all data exclusions (if any), all manipulations, and all measures in the study (see *Simmons, Nelson & Simonsohn, 2012*).

### Participants

Sample size estimation was based on the guidelines proposed for linear mixed-effects models (see the results section), according to which a minimum of 1,600 observations should be collected for each experimental cell (*Brysbaert & Stevens, 2018*). The minimum sample size requested for our experimental design (see the procedure section) was about 48 participants. The experiment was advertised among the student population *via* social media and email. We decided to stop data collection when no new responses were received, assuming that the minimum number of participants had been met. We closed data collection after about one week in which no new responses were recorded. The final sample consisted of 109 individuals (*Mean age* = 25 years, *SD* = 5.67, 38 males) who participated on a voluntary basis. All participants gave their informed consent through a specific online form. Data were collected between 26 March and 17 April 2021. The study was carried out according to the Declaration of Helsinki and was approved by the Ethics Committee for Psychological Research at the University of Padova (approval number: 3881).

### Stimuli, apparatus and procedure

The faces of 34 adult males, with a neutral expression, were extracted from the Karolinska Directed Emotional Faces (KDEF) database (*Lundqvist, Flykt & Ohman, 1998*). For each identity, there were two versions, namely one with the model showing the left side of his face (*i.e.,* the face appeared as oriented leftwards) and one with the model showing the right side of his face (*i.e.,* the face appeared as oriented rightwards; for some examples, see also Fig. 1; for KDEF codes, see also Appendix S1). During the experiment, half of the identities were constantly presented with the face oriented leftwards, and the other half with the face oriented rightwards. For each participant, the association between face identity and its orientation was randomly assigned to prevent any possible influence of perceptual differences among faces we did not consider.

   The task was developed taking inspiration from both the study by *Mendonça, Garrido & Semin (2020b)*, who presented participants with faces oriented leftwards and rightwards, and the study by *Jones et al. (2010)*, who observed a modulatory effect of dominance on social attention. The experiment was programmed through PsychoPy and delivered online through Pavlovia (*Bridges et al., 2020*). Each trial started with a black fixation cross (Arial font, 0.1° normalised unit; see also Fig. 1) for 500 ms, followed by the central picture of a task-irrelevant face (approximately 300 × 400 px). After a stimulus onset asynchrony

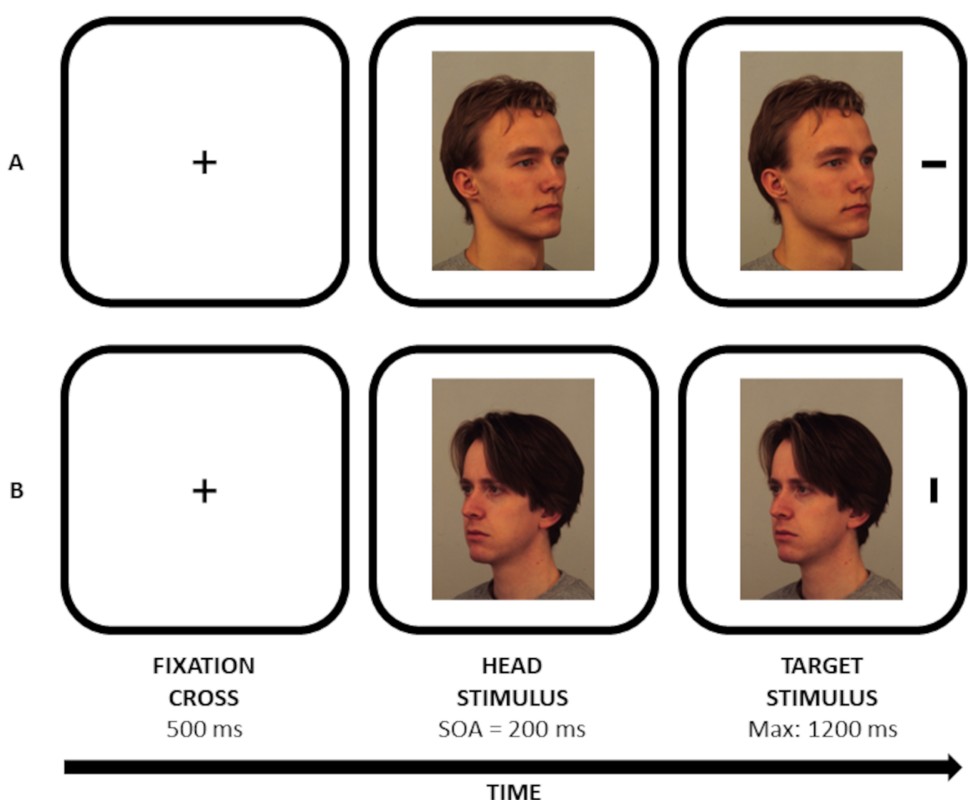

**Figure 1  Examples of stimuli (not drawn to scale) and trials employed in the experiment.** (A) shows an individual (AM08NEHR KDEF code) with the face oriented rightwards and a horizontal target line appearing on the right. (B) shows an individual (AM22NEHL KDEF code) with the face oriented leftwards and a vertical target line appearing on the right. Images (Copyright) KDEF stimulus set (The Karolinska Directed Emotional Faces; Lundqvist, Flykt, & Öhman, 1998).

(SOA) of 200 ms, a black target line (40 px width ×12 px height) appeared leftwards or rightwards (± 0.8 normalised units) with respect to the centre of the screen. In *Jones et al. (2010)*, the impact of dominance on gaze cueing was particularly evident at the 200-ms SOA, then decaying at longer SOAs. For this reason, a single SOA lasting 200 ms was employed here. Participants were instructed to discriminate the orientation of the line (*i.e.,* vertical *vs* horizontal) as quickly and accurately as possible by means of a key press (*i.e.,* f and k keys). A discrimination task was chosen for consistency with the works of *Mendonça, Garrido & Semin (2020b)* and *Jones et al. (2010)*. Participants were also told to maintain fixation in the centre of the screen for the entire duration of the trial. They were also asked to ignore face stimuli, as they were not informative with respect to the location of the target. The trial ended when a response was provided or 1,200 ms elapsed, whichever came first (see *Jones et al., 2010*). The association between the response key and the line was randomly assigned to the participants. In case of incorrect or missed responses, central visual feedback appeared for 500 ms (*i.e.,* the words 'NO' or 'TOO SLOW', respectively; Arial font, 0.1° normalised units). There was a practice block (10 trials) followed by an

experimental block (136 trials). Within the experimental block, all experimental conditions were presented an equal number of times in random order.

The main task was followed by a second task that aimed to assess the perceived level of dominance associated with the two types of faces. Following a procedure similar to that adopted by *Jones et al. (2010)*, participants were shown pairs of faces (one face oriented leftwards, the other face oriented rightwards), one appearing on the left side of the screen and the other one on the right side of the screen. Each facial stimulus used in the main task was randomly extracted and appeared only once (17 trials in total). The location of each face on the screen (*i.e.,* left or right) was also randomly determined. On each trial, participants were asked to decide which face appeared as 'more dominant' (that is, the one who, in a social situation, may be better able to guide and influence the other person). Responses were provided using two numerical keys (*i.e.,* 1 and 2). The two faces remained on the screen until a response was made and then a blank screen appeared for one second. Finally, participants were also asked to report their level of liberalism or conservatism using a five-point scale, with 1 = very liberal, 2 = liberal, 3 = middle-of-the-road, 4 = conservative and 5 = very conservative. This is a validated scale providing a reliable index of political temperament (see also, *e.g., Jost, 2006; Settle et al., 2010; Kanai et al., 2011*). We also opted for this tool because we wanted to present participants with a relatively short questionnaire, due to the online nature of the study. Responses were provided, with no time limits, by pressing the numerical key (*i.e.,* from 1 to 5) corresponding to the desired response. The whole experiment lasted about 15 min.

## RESULTS

### Data handling

Trials with a missing response were discarded (1.18% of trials), whereas trials with an incorrect response (9.87% of trials) were, for completeness, analysed separately. Correct trials with a latency less than 100 ms or greater than 3 SD from each participant's mean (calculated separately for each experimental condition) were considered outliers and discarded (0.96% of trials).

### Latencies and accuracy

Latencies of correct trials were analysed by adopting linear mixed-effects models implemented through the *lme4* package for R (*Bates et al., 2015*). For the sake of comparison with *Mendonça, Garrido & Semin (2020b)*, we considered as experimental factors face direction (2: leftwards *vs* rightwards) and target position (2: left *vs* right). The likelihood ratio test was employed for model comparison (ranging from the null model to the saturated model), indicating that the best model fitting the current data had face direction and target position as fixed effects, while the intercept for both participants and face identity, and the by-participant slope for the target position, were the random effects. This model was then analysed with an ANOVA implemented through the *lmerTest* package (*Kuznetsova, Brockhoff & Christensen, 2017*). Effect sizes were calculated following both the guidelines for linear mixed-effects models (hereafter labeled as '$d_{lme}$'; *Brysbaert & Stevens, 2018*) and a standard procedure (*i.e.,* not considering the random effects) for a more direct

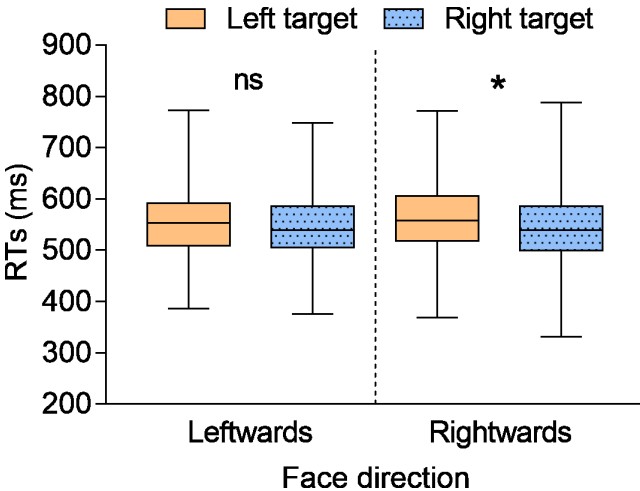

**Figure 2** **RTs as a function of the different experimental conditions.** Whiskers represent the minimum and maximum values. ns, not significant; *, $p < .05$.

comparison with previous studies on social attention. The main effect of face direction was not significant, $F(1, 65.3) = 0.312$, $p = .578$, $d_{lme} = 0.02$, $\eta^2_p < .001$, while the main effect of target position was significant, $F(1, 105.8) = 17.296$, $p < .001$, $d_{lme} = 0.11$, $\eta^2_p = .143$, due to smaller RTs for targets appearing on the right ($M = 546$ ms, $SE = 6.80$) than on the left ($M = 556$ ms, $SE = 6.59$). More importantly, the face direction × target position interaction was significant, $F(1, 12783.2) = 4.085$, $p = .043$, $d_{lme} = 0.06$, $\eta^2_p = .023$. This interaction was further analysed following the same approach adopted by *Mendonça, Garrido & Semin (2020b)*, in which the RTs for leftwards and rightwards targets were analysed separately for the two types of face, which also aligns with the standard approach used in social attention literature (see, *e.g.*, *Dalmaso, Castelli & Galfano, 2020*). Bonferroni-corrected planned comparisons were computed through the *lsmeans* package (*Lenth, 2016*). These showed that, while for faces oriented rightwards targets appearing on the right were responded to faster ($M = 545$ ms, $SE = 6.98$) as compared to targets appearing on the left ($M = 559$ ms, $SE = 6.77$; $p < .001$, $d_{lme} = -.107$, $d = -0.396$), for faces oriented leftwards no differences emerged between left ($M = 554$ ms, $SE = 6.77$) and right ($M = 547$ ms, $SE = 6.98$; $p = .116$, $d_{lme} = .045$, $d = 0.184$) targets (see also Fig. 2; see also Experiment 1 in (*Mendonça, Garrido & Semin, 2020b*), for a similar pattern of results).

Trials with an incorrect response were analysed using a mixed-effect logit model (*Jaeger, 2008*). The best model fitting the available data, according to the likelihood ratio test, had face direction (2: leftwards *vs* rightwards) and target position (2: left *vs* right) as fixed effects, while the intercept for participants and the by-participant slope for target position were the random effects. The only significant result was the main effect of target position, $b = -.29$, $SE = .091$, $p = .002$, $\eta^2_p = .035$, due to more errors for targets appearing on the left than on the right. No other significant results emerged ($ps > .203$).

## Perceived dominance

Data were analysed with a mixed-effect logit model, which is particularly adequate for dichotomous variables (*Jaeger, 2008*). In our case, the dichotomous response variable was codified in the following way. Trials in which participants selected the face placed on the right side of the screen were labelled '1', trials in which they selected the face placed on the left side as '0'. Then, we ran a model with the orientation (leftwards *vs* rightwards) of the face that appeared on the right side of the screen as a fixed effect, and participant as a random effect. No significant differences emerged, $b = .106$, $SE = .093$, $p = .257$, with a small odds ratio of 1.11 in favour of the face oriented leftwards. For completeness, we also conducted additional, explorative analyses in which the percentage of times right-oriented faces were judged as more dominant was used as a covariate in the linear mixed-effects model described above, but the results remained virtually identical.

## Relationship between the level of liberalism and conservatism and social attention

The responses on the five-point scale were polarised towards liberalism (19 participants responded '1' 42 responded '2' 35 responded '3', 11 responded '4', and 2 responded '5'). Responses to the political questionnaire were correlated with an overall index of the magnitude of the spatial cueing effect. This index was calculated following the standard approach used in social attention literature (*e.g.*, *Edwards et al., 2015*; *Carraro et al., 2017*) by subtracting the latencies of trials in which participants are generally faster (*i.e.*, the spatially-congruent trials) from the latencies of trials in which they are generally slower (*i.e.*, the spatially-incongruent trials). As for faces oriented leftwards, the mean latencies of targets appearing on the left were subtracted from the mean latencies of targets appearing on the right. The opposite computation was applied to faces oriented rightwards. A negative correlation emerged for faces oriented leftwards, $rho\ (109) = -.197$, $p = .040$, indicating that these stimuli elicited a weaker spatial cueing effect for participants with a more conservative political temperament. The correlation was not significant for faces oriented rightwards, $rho\ (109) = -.083$, $p = .393$ (see also Fig. 3).

## DISCUSSION

Social attention is an essential ability that allows us to successfully navigate within social contexts, establishing meaningful relationships with our conspecifics. Here, we explored whether faces oriented leftwards or rightwards could shape spatial cueing of attention differently. We asked participants to discriminate a peripheral target while a task-irrelevant face, oriented leftwards or rightwards, was presented at fixation. We also asked them to evaluate the perceived levels of dominance associated with facial stimuli and to report their level of liberalism or conservatism.

Our main results can be summarised as follows. First, we observed that, while for faces oriented rightwards targets appearing on the right were responded to faster as compared to targets appearing on the left, for faces oriented leftwards no differences emerged between left and right targets. This aligns with a previous work reporting a comparable pattern of results (*Mendonça, Garrido & Semin, 2020b*). Second, we found a negative correlation

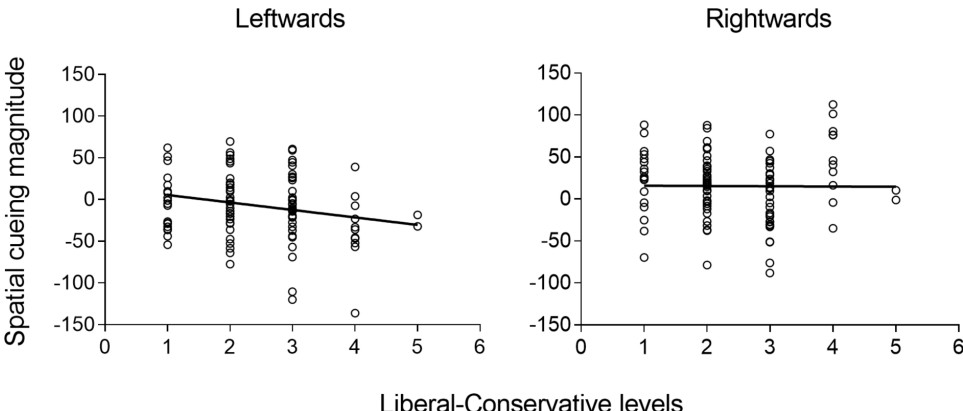

**Figure 3** **Correlations between spatial cueing magnitude and the level of of liberalism and conservatism.** Spatial cueing magnitude as a function of the level of liberalism and conservatism of the participants, represented separately for faces oriented leftwards (left panel) and rightwards (right panel).

between the level of conservatism expressed by participants and the magnitude of the spatial cueing effect elicited by the faces oriented leftwards. This provides additional support for the possible relationship between political temperament/affiliation and social orienting documented in previous studies (*e.g.*, *Dodd, Hibbing & Smith, 2011*; *Liuzza et al., 2011*; *Carraro et al., 2015*). For instance, *Dodd, Hibbing & Smith (2011)* reported a negative correlation between political temperament and the magnitude of social orienting elicited by a schematic face with an averted gaze (*i.e.,* the higher the degree of conservatism, the lower the orienting to gaze stimuli), likely reflecting the tendency of conservatives to be more individualistic and less permeable to others' influence. Third, we did not find supporting evidence for the notion that faces oriented rightwards were perceived as more dominant than faces oriented leftwards. In fact, the data provided by the task that aimed to collect an explicit measure of perceived dominance did not show any difference between the two orientations. This was unexpected and in contrast to previous works (*e.g.*, *Suitner, Maass & Ronconi, 2017*; *Mendonça, Garrido & Semin, 2020a*). Hence, our attempt to provide direct support for interpreting our results on social orienting in terms of a dominance perception effect failed. A possible explanation for this unexpected result may be related to the specific task we adopted, based on previous work on social attention (*Jones et al., 2010*), in which two facial stimuli were presented simultaneously, one on the left and one on the right side of the screen. We can tentatively suppose that, while this task may be optimal to compare two faces varying along an intrinsic physiognomic dimension, such as the degree of masculine or feminine traits (*Jones et al., 2010*), it could be less than ideal when the critical dimension associated with the two faces is purely spatial in nature (*i.e.,* a face oriented leftwards or rightwards). In other words, we suspect that the simultaneous presentation of two spatially-oriented faces, placed along the left–right axis, may have interfered with the hypothetical left–right spatial vector that would be implied in dominance evaluation. This possibility could be tested in future studies by directly comparing the performance when two faces or one single face are employed. In addition, the use of a Likert-like scale

to evaluate the perceived dominance associated with each face (see also *Suitner, Maass & Ronconi, 2017*; *Mendonça, Garrido & Semin, 2020a*) could be more appropriate than a dichotomous measure such as the one collected in our task.

Some limitations of the present study are related to the characteristics of our sample and facial stimuli. First, our sample was mainly composed of females. As there is evidence showing that the gender of participants can shape social attention (*i.e.,* females would tend to be more sensitive to social signals; see, *e.g.,* *Bayliss, di Pellegrino & Tipper, 2005*; *Dalmaso, Castelli & Galfano, 2020*), future studies could test the same number of females and males, to explore if gender is also involved in the phenomenon we explored. Second, most of the participants self-identified as liberals or centre-oriented. Even if this is common when students are tested (see, *e.g.,* *Woessner & Kelly-Woessner, 2020*), future studies could also try to get a more balanced sample in terms of political temperament, to increase the generalisability of the results. Regarding the facial stimuli, all of them belonged to male individuals. Although the gender of the face seems not involved in shaping social orienting (see *Bayliss, di Pellegrino & Tipper, 2005*), future studies could employ both male and female faces to increase the ecological validity of the results.

The presence of left–right spatial biases can be identified in several other domains other than social cognition. One of the most representative examples is provided by numerical cognition with the so-called Spatial–Numerical Association of Response Codes (SNARC) effect (*Dehaene, Bossini & Giraux, 1993*), according to which relatively small numbers are responded to faster with a key placed left (*vs* right) and relatively large numbers with a key placed right (*vs* left). This would reveal the tendency to represent numerical magnitude as a continuum ranging from left to right, at least in Western individuals. Interestingly, similar left–right effects have also been documented in other domains, such as time (*e.g.,* *Vallesi, Binns & Shallice, 2008*), size (*e.g.,* *Ren et al., 2011*) or weight (*e.g.,* *Dalmaso & Vicovaro, 2019*), suggesting a common tendency in the mental representation of magnitudes along space. Similar displacements have also been reported for valence, with negative-connoted stimuli that would be represented on the left side of space and positive-connoted stimuli on the right side of space (see, *e.g.,* *Holmes & Lourenco, 2011*; *Pitt & Casasanto, 2018*; *Dalmaso, Vicovaro & Watanabe, 2022*). The tendency to mentally represent dimensions of different natures within a spatial framework appears to be almost inevitable, and the results reported here suggest that it also embraces the domain of social attention (see also *Mendonça, Garrido & Semin, 2020b*).

According to some authors, the origins of these left–right spatial biases could be identified at a biological level, as they would arise from specific mechanisms related to hemispheric specialisation (*e.g.,* *Vallortigara, 2018*; *Felisatti et al., 2020*). This could explain why left–right spatial biases can be identified even among infants (*e.g.,* *de Hevia et al., 2017*) and animals such as chickens and apes (*Adachi, 2014*; *Rugani et al., 2015*). It is interesting to note that hemispheric specialisation could also impact social orienting mechanisms (*e.g.,* *Kingstone, Friesen & , 2000*; *Akiyama et al., 2006*; *Marotta, Lupiáñez & Casagrande, 2012*). Of relevance to the current work, *Marotta, Lupiáñez & Casagrande (2012)* tested healthy participants and found a reliable orienting of attention elicited by task-irrelevant eye-gaze stimuli presented centrally, but only when the target (*i.e.,* a letter) appeared in

the left visual field of the participants. This would likely reflect the fact that the attentional orienting response to eye-gaze stimuli would be governed by brain regions, deputed to face and eye-gaze processing, which would be mainly located in the right hemisphere (see also, *e.g.*, *Kingstone, Friesen & , 2000*). Even if this evidence could appear in contrast to that reported here, it should be noted that *Marotta, Lupiáñez & Casagrande (2012)* developed a task with the specific aim of testing gaze-mediated orienting of attention, and participants were presented with a central, schematic face, in which spatial information was provided by the two eyes only. In the current work, we used pictures of real faces and, more importantly, spatial information was provided by rotation of the whole head, which could explain the discrepancy between the two studies. Taken together, our and other works (*e.g.*, *Kingstone, Friesen & , 2000*; *Marotta, Lupiáñez & Casagrande, 2012*) seem to confirm that a combination of biological (*e.g.*, hemispheric specialisation), cultural (*e.g.*, reading/writing direction) and methodological (*e.g.*, cue type) factors would contribute to the emergence of spatial biases in social orienting.

## CONCLUSIONS

We explored whether faces oriented rightwards can elicit a stronger orienting of attention than faces oriented leftwards. The results aligned with this prediction and also showed that the magnitude of the spatial cueing effect elicited by faces oriented leftwards was associated with the level of liberalism and conservatism of the participants. These results confirm and extend previous work (*Mendonça, Garrido & Semin, 2020b*) and, more generally, offer new insights into the mechanisms governing social attention. However, unlike previous studies (*e.g.*, *Benjafield & Segalowitz, 1995*), we did not observe that faces oriented rightwards were perceived as higher in dominance than faces oriented leftwards.

Future studies could compare the performance of Western individuals with that of individuals with an opposite reading/writing direction (*e.g.*, Arabic) to investigate the impact of cultural habits on this phenomenon. One possible prediction is that, in Arabic individuals, a stronger orienting could emerge for faces oriented leftwards than rightwards. Furthermore, future studies could also employ different tasks (*e.g.*, target discrimination *vs* localisation), as there is evidence that in some contexts (*e.g.*, emotions; *Chen et al., 2021*) the nature of the task can influence orienting responses elicited by social stimuli. This could further probe the generalisability of the results observed here and in *Mendonça, Garrido & Semin (2020b)*.

### Funding
The authors received no funding for this work.

### Competing Interests
Mario Dalmaso is an Academic Editor for PeerJ.

## Author Contributions

- Mario Dalmaso conceived and designed the experiments, performed the experiments, analyzed the data, prepared figures and/or tables, authored or reviewed drafts of the article, and approved the final draft.
- Giacomo Fedrigo conceived and designed the experiments, performed the experiments, analyzed the data, prepared figures and/or tables, authored or reviewed drafts of the article, and approved the final draft.
- Michele Vicovaro conceived and designed the experiments, performed the experiments, analyzed the data, prepared figures and/or tables, authored or reviewed drafts of the article, and approved the final draft.

## Human Ethics

The following information was supplied relating to ethical approvals (i.e., approving body and any reference numbers):

Ethics Committee for Psychological Research at the University of Padova

## Data Availability

The raw data are available on OSF: Dalmaso, Mario, and Michele Vicovaro. 2023. "Gazing Left, Gazing Right: Exploring a Spatial Bias in Social Attention." OSF. April 26. doi: http://dx.doi.org/10.17605/OSF.IO/CJP9B.

## Supplemental Information

Supplemental information for this article can be found online at http://dx.doi.org/10.7717/peerj.15694#supplemental-information.

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
