# Peer review of "Gazing left, gazing right: exploring a spatial bias in social attention"

_PeerJ, doi:10.7717/peerj.15694_

## Round 0.1 · original submission · Major Revisions

Dear authors,

Please reply point by point to the reviewers' comments.

Reviewer 1 ·

Basic reporting

Overall, I thought this paper was well-written and easy to follow. The literature review (to my knowledge) is complete, and, in general, enough background/context has been provided to grasp the purpose of the study. The article is well-structured and contains the appropriate figures, and I thank the authors for providing a link to the raw data.

However, I'd like to see effect-size estimates included in the Results section, for both (1) the omnibus analysis and (2) the pairwise comparisons the authors performed to unpack the significant interaction. It looks like the interaction between face direction and target position was quite small, and I think effect sizes would be especially informative for interpreting the strength of this effect.

Experimental design

The research question seems meaningful and relevant given the background literature the authors presented. The experiment is well-designed, though I'd like to see further justification for some methodological decisions. Below are some specific suggestions.

- I think it would be good for the authors to explain why a replication of Mendonca et al. (2020) was warranted - is it because it's the only study that's investigated this question? I'm assuming that's the case, but it's always good to be explicit about the motivation of a study.

- Lines 112 - 114 were a little unclear to me in terms of the reasoning behind examining liberalism/conservatism. Rather than saying "a good candidate to further probe the link", maybe it would be clearer/more appropriate to say "to examine the possibility that the level of liberalism and conservatism is a factor that can influence the relationship between X and Y..." (or something along those lines).

- I think the authors could explain why they chose to use a discrimination task rather than, say, a localisation task. There's a bit of a discussion at the moment about how different tasks can lead to different cueing-effect magnitudes, and how different tasks can even determine whether or not the characteristics of a face (e.g., emotional expression) modulate gaze cueing. This makes me think that authors working in the gaze-cueing space should always justify why they've chosen the task they've used. See the reference below:
* * *
Chen, Z., McCrackin, S. D., Morgan, A., & Itier, R. J. (2021). The gaze cueing effect and its enhancement by facial expressions are impacted by task demands: direct comparison of target localization and discrimination tasks. Frontiers in Psychology, 12, 618606.
* * *
Validity of the findings

Here, I think the authors could do a little more to explain the importance of this question/research topic, both in the last paragraph of the Introduction and the first section of the Discussion. I found the topic personally interesting, but others might be left questioning why this topic is being investigated (e.g., why is it important to replicate Mendonca et al.'s (2020) finding?).

The raw data have been provided, the exclusion/screening criteria seem appropriate, and the linear mixed-effects analysis appears to have been competently performed. However, see my earlier point about the utility of reporting effect sizes.

I think the Discussion was well-balanced and the authors made the appropriate qualifications regarding their results (e.g., that there was no concrete evidence for rightwards-facing faces being perceived as more dominant than leftwards-facing faces). The clarity of the last sentence (lines 336 - 337) could be improved, though; specifically, it would be good to be clear that it's Arabic individuals that would be expected to show stronger orienting effects for leftwards (relative to rightwards) faces.

Additional comments

Overall, I think this manuscript makes an interesting contribution to the gaze-cueing literature, but would be improved with the abovementioned changes.

Reviewer 2 ·

Basic reporting

The paper is very well written and easy to follow. I would, however, advise more precision on the operational level in the literature review and postulating hypotheses. Further, in the discussion, the interpretations do not always follow from the results, and a lack of a basic effect is not properly discussed.

Experimental design

The design seems appropriate for the research question. The comments I had about the protocol are minor challenges or pertain more to the data - e.g., why was political affiliation measured with only one item, and what is the distribution of political affiliation amongst participants? Other issues pertain to results and reporting.

Validity of the findings

Some interpretations do not match the results. The interpreted greater effect of social attention of right vs. left face was not reported in the results, instead the effect the authors refer to is a difference in latency between congruent-incongruent target and face combination in one, but not the other, facial orientation. This seems at odds with the language in the discussion. Further, lack of a difference in dominance perception between leftward or rightward face orientation disallows interpretations of physiognomic traits influencing social attention.

Additional comments

Most of my review is in the attached document. Please, refer to it primarily. Thanks to the authors for a well written manuscript that was easy to review. I trust

Annotated reviews are not available for download in order to protect the identity of reviewers who chose to remain anonymous.

Reviewer 3 ·

Basic reporting

attached

Experimental design

attached

Validity of the findings

attached

Annotated reviews are not available for download in order to protect the identity of reviewers who chose to remain anonymous.

---

## Round 0.2 · Minor Revisions

I would like to thank the authors for engaging postively with the reviewers' comments. The manuscript has been significantly strengthened by the first revision.

However, reviewer 2 raises several important issues that all need to be satisfactorily addressed before the manuscript can be accepted for publication. In addition, reviewer 3 has indicated several more minor that would further strengthen the manuscript. I am confident that after addressing these concerns, the manuscript will be suitable for acceptance.

Reviewer 1 ·

Basic reporting

I am happy with the level of detail the authors have provided in their revised manuscript, especially their inclusion of effect sizes.

Experimental design

I appreciate the authors' clarification of different methodological decisions (e.g., the use of a discrimination task) and can see that further improvements to clarity have been made.

Validity of the findings

The authors have addressed my previous concern that the importance of their findings wasn't well-specified. With the inclusion of relevant effect sizes, I am also happy with the statistical aspects of the paper.

Reviewer 2 ·

Basic reporting

Overall very good with two outstanding caveats discussed in the attached letter.

Experimental design

Satisfactory methodology, with some challenges addressed by backing them up with literature standards.

Validity of the findings

One outstanding objection concerning interpretation. See attaced letter.

Additional comments

I am grateful to the authors for resubmitting the manuscript and addressing all the comments. I trust that your manuscript will be successful once the outstanding concerns are addressed.

Annotated reviews are not available for download in order to protect the identity of reviewers who chose to remain anonymous.

Reviewer 3 ·

Basic reporting

I would like to thank the Authors for considering my (and the other Reviewers') suggestions. I believe that the manuscript has improved significantly after the revision.

It now better reflects the results; the procedure used to analyse data is more transparent; and by removing the hypothesis on agentivity it is clearer. In my opinion it is now suitable for publication, I have only three minor points below.

Minor points
1. In the light of the changes made on the manuscript I suggest replacing "spatial cueing task" with "discrimination task" in the abstract.

2. Please, correct a typo on page 2 of the introduction femininised--> feminised.

3. Due to the lack of effect sizes provided by Mendonça et al., (no comparisons are possible), it will be more correct to provide a measure of the effect sizes of the significant results found in the linear mixed model analysis as it was the analysis used to test the hypothesis.

Experimental design

NA

Validity of the findings

NA

Additional comments

NA

---

## Round 0.3 · Minor Revisions

Many thanks for submitting a revised version of the manuscript.

However, at this stage I am not able to accept the manuscript as two important issues which were raised by Reviewer 2 in the last round have not been adequately addressed in your latest revision.

These issues concern the interpretation of your results in the Discussion section:
1) Reviewer 2 asked you to change the discussion from saying that the dominance effect is likely true, to saying that you did not observe the effect of dominance (which is true). This issue has not been satisfactorily addressed.

2) discussion about the (lack of) dominance perception requires a re-write, according to the comments below by Reviewer 2. This was not satisfactorily addressed in your revision.

These changes seems to be small enough to correct easily, however they are significant enough to be misleading about your results if left unaddressed. I will offer one last opportunity to appropriately revise your discussion in light of these two issues, so that the comments by Reviewer 2 can be fully addressed.

Reviewer 2 ·

Basic reporting

NA

Experimental design

NA

Validity of the findings

Regarding my two earlier major points:

1) changing 'reliable orienting response' to 'reliable social attention response' is not a great improvement. Your participants' attention/orienting was relatively reliable (discrimination was correct) across relevant conditions, but what the data suggests is a modulation (increase) of attention by presumably social factors in some, but not other conditions. 'Reliable' is misleading, and 'social attention response' is vague, because the whole formulation implies an attention or orienting response only in some conditions, which is not what was observed. The fact that a term resembles what is found in the literature does not mean it also fits your data or design.

2) The section of the discussion where the (lack of) dominance perception effect is discussed is remains inappropriate because no changes were made to it. I do not want to repeat my earlier comments, as they remain unaddressed. Currently, the paragraphs starting with line 307 and 330 are misleading. Do not 'tone down' the sentence, and do not 'make it speculative' - why should you speculate about your own data? Make the interpretation compatible with your data, and if you must discuss the literature's findings in a way that overshadows your data, then please state clearly that the two are in contrast. Rewrite those two paragraphs.

Additional comments

NA

---

## Round 0.4 · accepted · Accept

I am satisfied that the reviewers' comments have been appropriately addressed and that the manuscript is ready for publication.

Reviewer 2 ·

Basic reporting

Satisfactory, as in previous rounds.

Experimental design

Satisfactory, as in previous rounds.

Validity of the findings

Satisfactory, issue with interpretation in the discussion has been resolved.